# Nutritional Management and Role of Multidisciplinary Follow-Up after Endoscopic Bariatric Treatment for Obesity

**DOI:** 10.3390/nu14163450

**Published:** 2022-08-22

**Authors:** Anuradha Negi, Ravishankar Asokkumar, Rajesh Ravi, Gontrand Lopez-Nava, Inmaculada Bautista-Castaño

**Affiliations:** 1Department of Endocrinology, Raffles Hospital Specialist Center, 585 North Bridge Road, Singapore 188770, Singapore; 2Department of Gastroenterology and Hepatology, Singapore General Hospital, Singapore 169856, Singapore; 3Duke-NUS Graduate Medical School, Singapore 637551, Singapore; 4Bariatric Endoscopy Unit, HM Sanchinarro University Hospital, 28050 Madrid, Spain

**Keywords:** bariatric endoscopy, obesity, nutrition, weight loss, endoscopic sleeve gastroplasty, intragastric balloon

## Abstract

The prevalence of obesity has risen exponentially, and patients living with obesity suffer from its debilitating consequences. The treatment options for obesity have expanded significantly and include lifestyle changes, pharmacotherapy, endoscopic bariatric therapies (EBTs), and bariatric surgery. Endoscopic bariatric therapies comprise volume-reducing procedures such as endoscopic gastroplasty and gastric space-occupying devices such as intragastric balloons. Because of its minimally invasive nature and ease of delivery, EBTs are increasingly being adopted as a treatment option for obesity in several centers. These procedures mainly achieve weight loss by inducing early satiety and reducing meal volume. While the technical aspects of EBTs have been well explained, the nutritional management surrounding EBTs and the effectiveness of multidisciplinary team for maximizing weight loss is less described. There is considerable variation in post-EBT care between studies and centers. In this paper, we review the existing literature and share our experience on nutrition and the role of multidisciplinary management of obesity following EBT.

## 1. Introduction

Obesity is a chronic, relapsing, multifactorial, and neurobehavioral disease characterized by increased body fat and adipose tissue dysfunction [1]. The global prevalence of obesity has been steadily growing and has reached alarming levels in some areas of the world [2]. People living with obesity are at risk of developing debilitating comorbidities, including diabetes mellitus, hypertension, ischemic heart disease, non-alcoholic fatty liver disease, cancers, and stroke. Additionally, the quality of life and productivity of these patients are lower, and their health care expenditure is exceedingly high [3,4]. Currently, the treatment options for obesity extend from diet and lifestyle changes to pharmacotherapy and bariatric surgery. All of them have been shown to induce weight loss at varying levels, with surgery achieving more significant weight loss. However, the proportion of patients opting to undergo bariatric surgery has continued to remain low despite improvements in techniques and technologies [5]. Thus, a large treatment gap exists between the available therapies and patient preference toward obesity treatment.

Bariatric endoscopy represents a unique minimally invasive option developed based on principles learned from surgical techniques for obesity [6]. Broadly, they could be classified as gastric implantable devices such as intragastric balloons, restrictive gastric procedures such as endoscopic gastroplasty procedures, and therapies targeting the small bowel. Multiple other therapies targeting different mechanisms are in development [7,8]. However, among the available options, intragastric balloons (IGBs) and endoscopic gastroplasties (EGs) are the widely utilized procedures for obesity treatment. Pooled data from randomized controlled trials showed that percent total body weight loss (%TBWL) improved at 12 months for patients who received IGB therapy vs. those undergoing lifestyle modification alone (mean difference [MD], 4.42%; 2.90–5.95%) [9]. Likewise, the pooled rate of %TBWL with endoscopic sleeve gastroplasty at 12 months was 17.1% (95% CI: 15.1–19.1) [10]. The observed superior weight loss results with bariatric endoscopy treatment options have led to their inclusion in the guidelines by the American Gastroenterological Association, American Association of Clinical Endocrinologists (AACE), The Obesity Society (TOS), and the American Society for Metabolic and Bariatric Surgery (ASMBS) [10,11].

The evidence supporting endoscopic bariatric therapies (EBTs) as a weight-loss option has grown over the years. Most of them deliver EBTs within a multidisciplinary bariatric program rather than in isolation. However, we could still observe a variation in weight loss results across different studies and centers over time. A considerable difference exists in the lifestyle intervention, dietary recommendation, and follow-up intervals between programs. There is no standardized approach to patients after bariatric endoscopy treatment, nor is there guidance to physicians caring for such patients. Therefore, it is relevant to understand what makes patients undergoing EBTs achieve weight loss. In this article, we share the evidence, our nutrition experience, and the importance of multidisciplinary follow-up after EBT.

## 2. Endoscopic Bariatric Therapies and Nutritional Care

The backbone for success of any bariatric program is designing an intervention program that involves (a) nutritional intervention in the form of a low-calorie healthier diet, (b) physical activity planning, (c) behavior changes, and (d) overcoming psychological barriers to achieve and maintain long-term weight loss. The application and implementation of such interventions become synergistic when they are aligned with the function and physiological changes associated with EBTs.

## 3. Intragastric Balloons

There are several designs of intragastric balloons (Figure 1A,B) based on its implantation technique (swallow vs. endoscopic placement), filling substance (fluid vs. air), duration of placement (4 months vs. 6 months vs. 12 months), and adjustability. In general, fluid filled IGBs (400–700 mL) are commonly used and left in place for at least 6–12 months [12,13]. The IGB functions by restricting the gastric volume and through its effect on gastric mechanoceptors, induces a feeling of fullness after a meal. Additionally, the IGB reduces the gastric emptying for solids significantly but not for liquids [14]. The effect of IGBs on hunger hormone ghrelin is variable [14,15]. In the initial phase after IGB placement, patient would experience intolerance symptoms such as vomiting and abdominal pain to varying degrees for a period of 7–10 days. Thus, early nutrition and the transition in diet should be adjusted to patients’ adaptation to IGBs.

## 4. Endoscopic Gastroplasty

The frequently performed EG procedures include endoscopic sleeve gastroplasty (ESG) using the overstitch device (Figure 2A,B) and the modified primary obesity surgery endoluminal procedure (POSE-2.0). The ESG and POSE-2.0 procedure aim to shorten and reduce the size of stomach using specialized transmural sutures [16,17]. Both procedures leave the fundus of the stomach intact. Similar to IGBs, patients experience a feeling of fullness immediately after a meal and the meal portion sizes are significantly reduced. Physiological studies have shown slowing of gastric emptying to solids after ESG and a 59% reduction in calorie consumption to achieve maximum fullness [18]. We also found that ESG prevents compensatory rise in ghrelin levels and prevent rebound hunger without weight loss. It also improves insulin resistance and secretory patterns but has no impact on hind-gut hormones (GLP-1 and Peptide-YY) [19]. Patients have rapid recovery with minimal post-procedure symptoms and, unlike bariatric surgical procedures, dumping syndrome is less of a concern.

## 5. General Nutritional Principles

### 5.1. Pre-EBT Evaluation

In our practice, we conduct a baseline nutritional assessment similar to the ASMBS nutritional guidelines [11,20]. We perform anthropometric measurements and obtain weight histories such as failed weight loss attempts, current co-morbidities, food allergies, eating disorders, current/past psychiatric diagnosis, alcohol/tobacco use, dietary intake, physical activity level, and psychosocial factors such as motivation level, readiness to change, stress and coping mechanism. This information is crucial for adaptation to eating habits after the procedure and addresses individual barriers to weight loss success.

Micronutrient deficiencies are frequently encountered among patients with obesity due to overconsumption of low-nutrient, high-calorie foods that lack nutrient densities before EBTs [20]. Deficiency in iron, B-vitamins, vitamin D, and folate has been reported. Malabsorptive procedures such as Roux -en -Y gastric bypass are known to exacerbate micronutrient depletion of vitamin B12 and other B vitamin complex, iron and calcium. Restrictive procedures, such as EBTs, do not induce malabsorption. However, poor food choices, food intolerance and restricted portion size can contribute to micronutrient deficiency. Recognition and correcting the micronutrient deficiencies before is considered optimal to prevent post-EBT deficiency. We prescribe a multivitamin-mineral supplement to meet the daily requirements.

Among patients undergoing bariatric surgery, there are variable data on the role of preoperative weight loss using a restricted-calorie diet on operative and mortality outcomes. However, the potential advantage of such a recommendation is still controversial. The recent position statement from the American Society for Metabolic and Bariatric Surgery concluded that there is a lack of evidence to support insurance-mandated (time-based) preoperative weight loss as there is no association with morbidity or mortality [21]. Akin to this recommendation, routine pre-EBT weight loss, especially before EGs, is not suggested. Nonetheless, providing a time to assess patients’ motivation and willingness to adhere to nutrition advice and follow-up could be considered.

### 5.2. Peri-Procedure

The initial period after IGB or EG therapy is mainly focused on symptom control, particularly reducing the episodes of vomiting, retching, and abdominal pain. Many of these patients have metabolic illnesses and diabetes mellitus, and monitoring and optimizing their glycemic control takes precedence. The gastric symptoms are more pronounced with IGBs than EGs as the degree of gastric stasis is greatest during the earlier weeks. Nunes et al. recommended taking a cold liquid diet such as coconut water or lime juice during the first three days to allow for gastric adaptation and symptom control after IGB therapy and progressively increasing the calories of the liquid diet to 800 Kcal until Day 14 [22]. Kotzampassi and Lopez-Nava et al., also recommended initiating and maintaining patients on a liquid diet for the first 2 weeks [23,24]. In our practice, we have adopted a similar diet protocol and standardized it for IGBs and EGs (Table 1).

#### 5.2.1. Stage 1

This phase starts on the day after surgery and lasts for 1 week. The patient is maintained on a liquid diet such as sugar-free strained juices, strained vegetable soup, sugar-free liquid yogurt, and sugar-free isotonic drinks. It is essential to take small volumes at frequent intervals to prevent rapid gastric distension and minimize intolerance. The approximate calorie intake during this period is around 400 Kcal/day. Careful attention must be paid to the hydration status, and ad libitum water intake should be reinforced.

#### 5.2.2. Stage 2

The second phase starts when the gastric distress symptoms have improved and the patient can tolerate more fluids. Typically, this occurs during the second week after EBTs. During this stage, we increase the volume of the liquid diet and introduce protein shakes, pureed vegetables, and egg whites. The approximate calorie intake during this period is around 400–600 Kcal/day. Throughout the stages, good hydration and frequent meal intake in smaller volumes are stressed.

#### 5.2.3. Post-Procedure

We monitor patients’ progress at regular intervals during the initial phase, similar to the bariatric surgical recommendations [11]. The follow-up assessment should involve a multidisciplinary team interested in bariatric nutrition and care. We assess for symptoms, food intolerance, and psychological status as halitosis, vomiting, constipation, low mood, and insomnia are frequently encountered. In the absence of significant symptoms, we escalate the diet plan.

#### 5.2.4. Stage 3

Following the 2 weeks of liquid diet, which allows healing and cicatrization after EGs, and gastric adaptation with IGB, therapies, we progressively increase the consistency of the food. In the third week, in addition to liquids and protein shakes, we introduce a more pureed diet, egg protein, and add healthy fat. The calorie intake is approximately 600–800 Kcal.

#### 5.2.5. Stage 4

During this stage (4–5th week), we transition the diet to more solid food and normal consistency. We recommend a low-calorie diet and strictly encourage healthy eating habits. As a behavior change, we encourage patients to maintain a meal diary and actively engage in physical activity. We increase the calorie intake from 800 to 1200 Kcal and suggest to convert protein shakes to semi-solid or solid protein sources.

#### 5.2.6. Healthy Eating Practice

The peri and post-operative periods provide ample opportunity to re-educate the patient on long-term nutritional goals and assist in modifying the dietary habits. Guiding their daily meal’s micro and macronutrient composition may enable them to achieve and maintain weight loss in the long term (Figure 3).

In the first month after EBTs, the patient might experience significant weight loss, possibly attributed to muscle and fat-free mass loss [25]. Similar to the current bariatric surgical guidelines, we recommend a higher intake of proteins 60–80 g/day or up to 1.5 g/kg/day to prevent lean body mass [11,26]. A randomized study by Oppert et al. showed that the high-protein diet coupled with supervised strength training could overcome the loss in muscle strength after bariatric surgery [27]. The protein-rich diet exerts a higher thermogenic effect and leads to an excess energy expenditure to digest and store compared to carbohydrates and fats. Additionally, a high-protein diet promotes prolonged satiety and controls appetite by stimulating the secretion of glucagon-like peptide-1 (GLP-1) and peptide YY [28,29].

Attaining a negative energy balance (~300–500 Kcal) is crucial for sustained weight loss. We emphasize more on the quality of diet than the quantity. The DIETFITS randomized clinical trial did not show a significant decrease in weight loss between low-fat and low-carbohydrate diets at 12 months [30]. However, the quality of the diet- low intake of processed foods and added sugars, high intake of fruit, vegetables, and whole-grain products determined the weight loss in obese patients. As per the carbohydrate-insulin model of obesity, carbohydrates elevate insulin secretion leading to fat storage in adipose tissue; therefore, we advise our patients to limit processed foods, carbonated beverages, fruit juices and added sugars to less than 10% of daily calorie intake [31]. We routinely prescribe dairy-based products such as sugar-free yogurt and low-fat milk as they promote fat loss. We recommend patients to have 4–5 small meals during the day, eat slowly, chew thoroughly and stop eating once they reach satiety. We advocate water consumption of 30 mL/kg/day and avoid taking food and beverages at the same time. It has been shown that having 4 to 5 meal frequency was associated with a good weight-loss-response at 3 and 12 months after EBTs [32].

### 5.3. Dietary Planning

Generally, it is challenging to lose weight without achieving a negative energy balance, which is determined by energy intake and expenditure. The energy balance is regulated by a complex interaction of endocrine, metabolic, and nervous system signals to control food intake in response to the body’s dynamic energy needs and environmental influences [31]. We expend energy through resting metabolic rate—the energy required for the body’s functioning at rest; physical activity; and thermogenic effect of food—the energy needed for digestion, absorption, and storage of nutrients in the food. Change in body weight occurs when energy expenditure is higher than intake over a given period of time, resulting in loss of body fat mass. When designing a dietary intervention, an individualized diet that achieves a state of negative energy balance should be prescribed. Several dietary approaches are available based on the inclusion and restriction of different food components to achieve and maintain weight loss (Figure 4). These include modification of macronutrients formulation of the food (low-carbohydrate diet; low-fat diet; high-protein diet); inclusion of diverse food groups (Mediterranean diet—based on rich plant-based food and moderation of refined grains, red meat, and dairy); restriction of specific foods (paleo diet, vegan diet, and gluten-free diet) and time-restricted eating [33]. A recent network meta-analysis of randomized trials on 14 popular macronutrient diets showed that most diets resulted in modest weight loss and improved blood pressure over six months [34]. At 12 months, weight reduction diminished, and blood pressure improvements largely disappeared. Likewise, a randomized study comparing time-restricted eating to a daily low-calorie diet showed no difference in reduction in body weight, body fat, or metabolic risk factors in patients with obesity [35]. Thus, to promote long-term adherence, any dietary intervention should take into consideration patients’ food preferences, cultural background, and food availability.

There is a scarcity of literature surrounding type of meal planning for the long term after EBTs. The limited evidence focuses on use of either the Mediterranean diet or the high-protein diet [36]. The Mediterranean diet (carbohydrates—53%, proteins—22%, and fats—25%) is characterized by an abundance of plant-based foods, a moderate intake of fish and dairy, a low intake of red meat, and the use of extra virgin olive oil as the main source of dietary fat. The anti-inflammatory and anti-oxidant property of the Mediterranean diet has been postulated to promote weight loss and improvement on comorbidities. Short-term studies (3 months) have demonstrated superiority of the Mediterranean diet over the high-protein diet in inducing weight loss in obesity. However, long-term studies in EBTs are lacking.

The high-protein diet is the frequently subscribed diet post-EBTs. The protein-based bariatric plate model by Cambi et al. describes the macronutrient and micronutrient composition of daily meals to promote and maintain weight loss over the long-term after EBTs (Figure 5). The model recommends 50% calcium and iron-rich protein, 30% vitamins, minerals, and fibers, represented by fruits and vegetables, and 20% whole carbohydrates with a low glycemic index [37]. Lipids in the form of canola or olive oil were suggested. The high-protein and fiber-rich diets are thought to be associated with improved satiation, satiety, and reduced food intake. In our practice, we use a combination of the Mediterranean and high-protein diets to promote weight loss based on patient preference.

Optimal micronutrient levels are essential for various biological processes such as fat and carbohydrate metabolism, thyroid function, and glucose homeostasis. Their deficiency is frequent in obesity and could be aggravated after bariatric surgery due to malabsorption. Conversely, EBTs do not induce malabsorption as the anatomy, and the function of the gastrointestinal tract are unaltered. Micronutrient deficiencies were only reported in a small subset of patients with EBTs, and supplementation in at-risk patients could be considered [38].

At times, the nutrition management information after EBTs could overwhelm patients, leading to dropouts and non-compliance. To simplify the daily adherence to healthy eating practices, a visual description of how to plan the meal and design its composition would be helpful. The bariatric eating plate model may be an easy tool for patients to prepare their meals [37].

### 5.4. Role of Follow-Up Adherence and Weight Loss after EBTs

Obesity is a multifaceted problem, and health care delivery must adhere to the core principles of the chronic care model, which emphasizes long-term commitment, patient-centered intervention, using evidence-based protocol-driven care, and delivery of treatment within a multidisciplinary team [39]. Having a multidisciplinary unit comprising gastroenterologists, bariatric surgeons, obesity medicine physicians, nutritionists, physiotherapists, pharmacists, and case managers would facilitate addressing patients’ unique care and concerns, and personalize the treatment (Table 2). Several studies have emphasized the importance of adherence to follow-up with a multidisciplinary bariatric team to optimize weight loss results post-surgery [40]. Sarwer et al. showed that patients who received dietary counseling achieved more significant weight loss after bariatric surgery than those who received standard post-operative care at six months (26.1% vs. 32.5%) [41]. Nonetheless, adherence to follow-up remains remarkably challenging, and the attrition rate reported could be as high as 40–62% after 2 years among the bariatric surgical patients [42,43].

Although EBTs are less invasive than surgery, the lessons learned from bariatric surgery on the role of adherence still apply to EBTs. The Brazilian IGB practical guidelines recommend dietitian support after IGB placement as essential [44]. Amando et al. in a retrospective study involving 159 IGB patients, showed more significant benefit in weight loss and BMI reduction with psychological follow-up and exercise intervention within a multidisciplinary team [45]. Gemma et al. prospectively studied the role of adherence to the multidisciplinary interventional program after EBTs (IGB and POSE). They showed that patients who adhere to healthy lifestyle habits exceeded the weight loss threshold of bariatric surgery (EWL > 50%). At 12 months, good weight-loss responders lost 67.5% EWL, while poor weight-loss responders gained 5.7% of the EWL that they had lost at the 6 month follow-up. Irrespective of the type of EBT, patients who adhered to the MDT recommendation had 4.37-fold greater odds [95% CI: 2.19, 8.88] of being a good weight-loss responder [32]. There is considerable variation in the follow-up frequency and interval between the studies.

In our practice, we follow-up patients weekly or biweekly post-procedure. When their condition stabilized, we extended the visits to once a month. We recommend achieving 20–24 clinic visits over one year, irrespective of the procedure type. In our study, involving 962 patients who underwent EBTs, we found a higher percentage of follow-up attendance significantly predicted %TBWL at 1 year [46]. We also observed patients who underwent endoscopic gastroplasties adhered to follow-up at 1 year, further reiterating the importance of adherence to achieve successful weight loss. There is no one-size-fits-all approach to treating obesity. We believe introducing new, appealing, patient-centered interventions, involving family members, and appreciating the benefits they have already achieved may promote long-term adherence to weight maintenance efforts.

## 6. Conclusions

Significant progress has been achieved in obesity management in recent decades, and patients now have various treatment options to choose from to meet their needs and weight loss goals. Endoscopic bariatric procedures have become an attractive and effective therapeutic approach for obesity and are widely adopted in clinical practice. Nonetheless, adhering to the fundamental tenets of obesity management is essential to realize the benefit of EBTs. Nutrition is still the cornerstone, and designing a nutritional plan that is synergistic with the function of EBTs would enhance weight loss. Likewise, the multidisciplinary team represents the pillars of a bariatric endoscopic program. Working together cohesively with a common focus and standardized evidence-driven protocols would promote treatment adherence and success.

## Figures and Tables

**Figure 1 nutrients-14-03450-f001:**
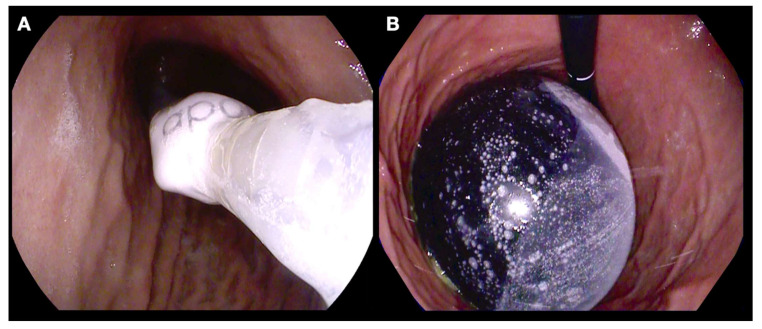
Endoscopic placement of a fluid filled intragastric balloon. (**A**) Endoscopic guided insertion of a gastric balloon catheter; (**B**) an IGB filled with 650 mL of fluid.

**Figure 2 nutrients-14-03450-f002:**
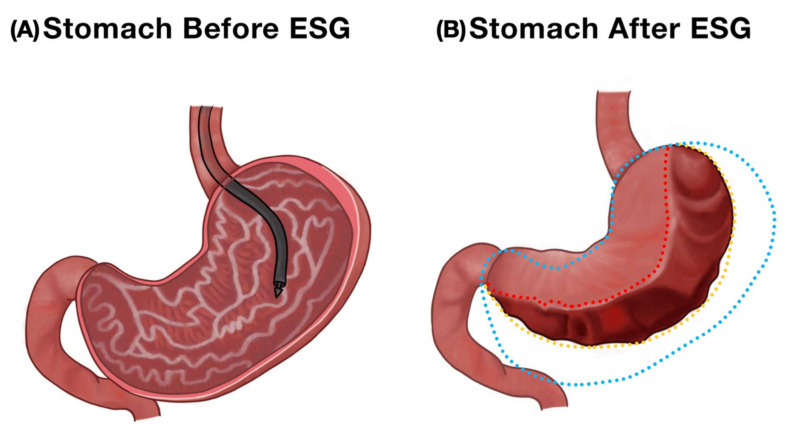
Endoscopic appearance of the stomach before and after endoscopic sleeve gastroplasty. (**A**) Normal stomach. (**B**) A reduced gastric volume after transmural suturing in ESG.

**Figure 3 nutrients-14-03450-f003:**
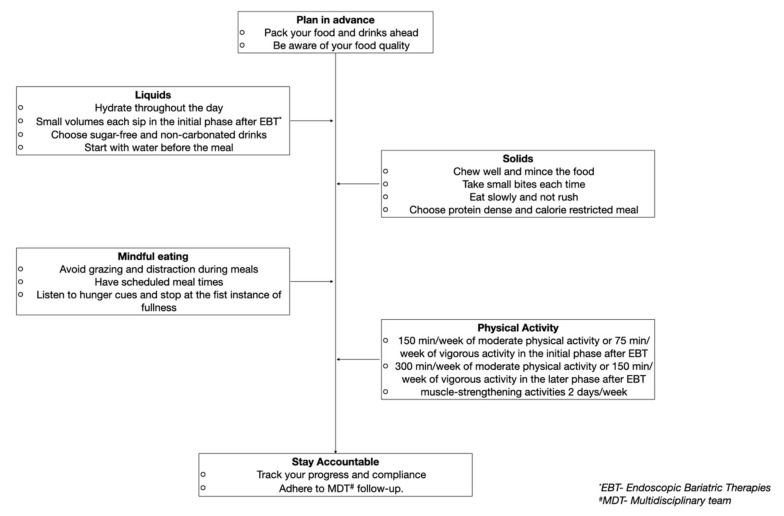
Example of a patient centered recommendation post-endoscopic bariatric therapies to achieve and maintain long-term weight loss.

**Figure 4 nutrients-14-03450-f004:**
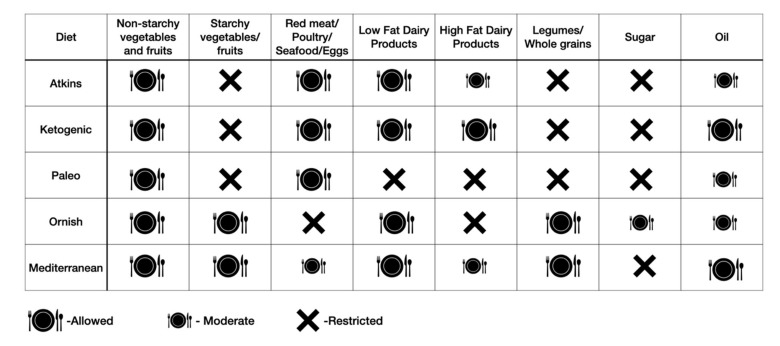
Popular dietary approaches for obesity based on inclusion and exclusion of specific food groups.

**Figure 5 nutrients-14-03450-f005:**
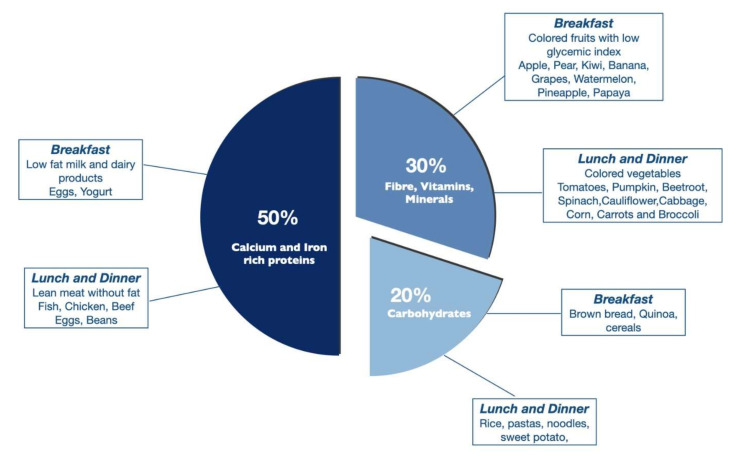
Example of the bariatric plate model based on macronutrient composition with breakfast and lunch options.

**Table 1 nutrients-14-03450-t001:** Nutrition plan in the immediate phase after endoscopic bariatric therapies.

Meal Plan	Meal Type	Estimated Calorie	Examples
Stage 1 (1st week)	Liquid diet	400 kcal/day	Sugar free strained juices of pear, apple or peach, strained vegetable soup, sugar free isotonic drinks
Stage 2 (2nd week)	Pureed diet	600 kcal/day	Vegetable puree such as potato, carrots, yogurt, cooked egg whites
Stage 3 (3rd week)	Mechanically altered soft diet (chopped, ground, mashed or pureed)	600–800 kcal/day	Fruit compote, cooked egg whites with olive oil
Stage 4 (4–5th week)	Transition to regular texture diet	600–800 kcal/day	Vegetable puree with chicken, meat or fish, cooked egg white or omelete, fresh cheese

**Table 2 nutrients-14-03450-t002:** Multidisciplinary care team and their role in endoscopic bariatric therapies.

Member	Role
Physician	Lynchpin of a multidisciplinary team Discuss weight and lifestyle Manage comorbidities and exclude secondary causes of obesity Gastroenterologist for endoscopic bariatric therapy Surgeon for bariatric surgery
Dietician	Laying the foundation for dietary change Assessment of patient’s diet Plan simple effective diet strategies Propose tailored dietary regime for weight loss before and after EBTs
Clinical psychologist	Mentally preparing the patient Identify psychosocial factors and barriers contributing to obesity Motivational interviewing and goal setting Managing expectations Cognitive restructuring Relapse prevention
Physical therapist	Integrating physical activity into healthy lifestyle Assessment of patient’s physical condition Plan realistic and achievable goals Propose tailored exercise regime for weight loss before and after EBTs
Specialist nurse/case manager	Keeping the patient involved To educate patient pre- and post-EBT To assist in tracking patient’s progress To assist communication between team members Scheduled counselling sessions
Pharmacist	To work with physicians on optimizing pharmacological therapy for weight loss

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
