# Peer review of "Nutritional Management and Role of Multidisciplinary Follow-Up after Endoscopic Bariatric Treatment for Obesity"

_nutrients, 2022, doi:10.3390/nu14163450_

Round 1

Reviewer 1 Report

The manuscript by Negi A, et al., titled as "Nutritional Management and Role of Multidisciplinary Follow- up After Endoscopic Bariatric Treatment for Obesity" is a review article, authors have reviewed methods involved in bariatric treatment, post-treatment followup methods and nutritional interventions.

Here are my comments to authors:

1. The manuscript not meeting the satisfying level of level of "title"  "Nutritional Management and Role of Multidisciplinary Follow- up" it is lacking information on nutritional management. 

2. Explain in detail what is mean by MDT followup, accountability? and how a patient can follow up?

3. authors have mentioned (line 190negative energy balance, write more on it, and give details/recent literature about diet/food to achieve it. 

4. Fig. 4 is good, but missing a good placement. try to write more on it and put it before conclusion.

5. I have missed the supplementary data, I couldn't find it in this manuscript. 

please provide supplementary files.

Author Response

Prof. Maria Luz Fernandez

Editor-in-Chief

Nutrients Journal

Department of Nutritional Sciences,

University of Connecticut, Storrs, CT 06269, USA

Manuscript ID: nutrients-1804837

July 24, 2022

Dear Editor and Reviewers,

Thank you very much for the reviews and comments. We have amended the manuscript as suggested.

Reviewer 1

1) The manuscript not meeting the satisfying level of level of "title"  "Nutritional Management and Role of Multidisciplinary Follow- up" it is lacking information on nutritional management.

We have amended the manuscript and added more information on nutrition.

General Nutritional Principles

Pre-EBT evaluation

In our practice, we conduct a baseline nutritional assessment similar to the  ASMBS nutritional guidelines [11, 20]. We perform anthropometric measurements and obtain weight histories such as failed weight loss attempts, current co-morbidities, food allergies, eating disorders, current/past psychiatric diagnosis, alcohol/tobacco use, dietary intake, physical activity level, and psychosocial factors such as motivation level, readiness to change, stress and coping mechanism. This information is crucial for adaptation to eating habits after the procedure and addresses individual barriers to weight loss success.

Micronutrient deficiencies are frequently encountered among patients with obesity due to overconsumption of low nutrient, high-calorie foods that lack nutrient densities before EBT [20]. Deficiency in iron, B- vitamins, vitamin D, and folate has been reported. Malabsorptive procedure like Roux -en -Y gastric bypass is known to exacerbate micronutrient depletion of vitamin B12 and other B vitamin complex, iron and calcium. Restrictive procedures, such as EBT,  do not induce malabsorption. However, poor food choices, food intolerance and restricted portion size can contribute to micronutrient deficiency Recognition and correcting the micronutrient deficiencies before is considered optimal to prevent post-EBT deficiency. We prescribe a multivitamin-mineral supplement to meet the daily requirements.”

“Attaining a negative energy balance (~300-500Kcal) is crucial for sustained weight loss. We emphasize more on the quality of diet than the quantity. The DIETFITS randomized clinical trial did not show a significant decrease in weight loss between low fat and low carbohydrate diets at 12 months [30]. However, the quality of the diet- low intake of processed foods and added sugars, high intake of fruit, vegetables, and whole-grain products determined the weight loss in obese patients. As per the carbohydrate-insulin model of obesity, carbohydrates elevate insulin secretion leading to fat storage in adipose tissue, therefore we advise our patients to limit processed foods, carbonated beverages, fruit juices and added sugars to less than 10% of daily calorie intake [31].”

“Dietary Planning

Generally, it is challenging to lose weight without achieving a negative energy balance, which is determined by energy intake and expenditure. The energy balance is regulated by a complex interaction of endocrine, metabolic, and nervous system signals to control food intake in response to the body's dynamic energy needs and environmental influences [31]. We expend energy through resting metabolic rate- the energy required for the body's functioning at rest; physical activity; and thermogenic effect of food- the energy needed for digestion, absorption, and storage of nutrients in the food. Change in body weight occurs when energy expenditure is higher than intake over a given period of time, resulting in loss of body fat mass. When designing a dietary intervention, an individualized diet that achieves a state of negative energy balance should be prescribed. Several dietary approaches are available based on the inclusion and restriction of different food components to achieve and maintain weight loss (Figure-4). These include modification of macronutrients formulation of the food (low carbohydrate diet; low-fat diet; high protein diet); inclusion of diverse food groups (Mediterranean diet- based on rich plant-based food and moderation of refined grains, red meat, and dairy); restriction of specific foods (Paleo diet, Vegan diet, Gluten-free diet) and time restricted eating [33]. A recent network meta-analysis of randomized trials on 14 popular macronutrient diets showed that most diets resulted in modest weight loss and improved blood pressure over six months [34]. At 12 months, weight reduction diminished, and blood pressure improvements largely disappeared. Likewise, a randomized study comparing time restricting eating to a daily low calorie diet showed no difference in reduction in body weight, body fat, or metabolic risk factors in patients with obesity [35]. Thus, to promote long-term adherence, any dietary intervention should take into consideration the patients' food preferences, cultural background, and food availability.

There is scarcity of literature surrounding type of meal planning for long term after EBTs. The limited evidence focuses on use of either Mediterranean diet or high protein diet [36]. The Mediterranean diet (carbohydrates-53%, proteins-22%, and fats-25%) is characterized by an abundance of plant-based foods, a moderate intake of fish and dairy, a low intake of red meat, and the use of extra virgin olive oil as the main source of dietary fat. The anti-inflammatory and anti-oxidant property of the Mediterranean diet has been postulated to promote weight loss and improvement on comorbidities. Short term studies (3 months) have demonstrated superiority of Mediterranean diet over high protein diet in inducing weight loss in obesity. However, long term studies in EBT are lacking.

The high protein diet is the frequently subscribed diet post-EBTs. The protein-based bariatric plate model by Cambi et al. describes the macronutrient and micronutrient composition of daily meals to promote and maintain weight loss over the long term after EBT (Figure-5). The model recommends 50% calcium and iron-rich protein, 30% vitamins, minerals, and fibers, represented by fruits and vegetables, and 20% whole carbohydrates with a low glycemic index [37]. Lipids in the form of Canola or olive oil were suggested. The high protein and fibre rich diet are thought to be associated with improved satiation, satiety, and reduced food intake. In our practice, we use a combination of Mediterranean and high protein diets to promote weight loss based on patient preference.”

2) Explain in detail what is mean by MDT followup, accountability? and how a patient can follow up?

We have added the following to the manuscript “Obesity is a multi-faceted problem, and health care delivery must adhere to the core principles of the chronic care model, which emphasizes long-term commitment, patient-centered intervention, using evidence-based protocol-driven care, and delivery of treatment within a multidisciplinary team [39]. Having a multidisciplinary unit comprising gastroenterologists, bariatric surgeons, obesity medicine physicians, nutritionists, physiotherapists, pharmacists, and case managers, would facilitate addressing patients' unique care and concerns, and personalize the treatment (Table-2).”

“There is no “one-size-fits-all approach to treating obesity. We believe introducing new, appealing, patient-centered interventions, involving family members, and appreciating the benefits they have already achieved may promote long-term adherence to weight maintenance efforts.” We have included a Table on the components of a multidisciplinary team.

3) authors have mentioned (line 190) negative energy balance, write more on it, and give details/recent literature about diet/food to achieve it.

We added to the manuscript “Generally, it is challenging to lose weight without achieving a negative energy balance, which is determined by energy intake and expenditure. The energy balance is regulated by a complex interaction of endocrine, metabolic, and nervous system signals to control food intake in response to the body's dynamic energy needs and environmental influences [31]. We expend energy through resting metabolic rate- the energy required for the body's functioning at rest; physical activity; and thermogenic effect of food- the energy needed for digestion, absorption, and storage of nutrients in the food. Change in body weight occurs when energy expenditure is higher than intake over a given period of time, resulting in loss of body fat mass. When designing a dietary intervention, an individualized diet that achieves a state of negative energy balance should be prescribed. Several dietary approaches are available based on the inclusion and restriction of different food components to achieve and maintain weight loss (Figure-4). These include modification of macronutrients formulation of the food (low carbohydrate diet; low-fat diet; high protein diet); inclusion of diverse food groups (Mediterranean diet- based on rich plant-based food and moderation of refined grains, red meat, and dairy); restriction of specific foods (Paleo diet, Vegan diet, Gluten-free diet) and time restricted eating [33]. A recent network meta-analysis of randomized trials on 14 popular macronutrient diets showed that most diets resulted in modest weight loss and improved blood pressure over six months [34]. At 12 months, weight reduction diminished, and blood pressure improvements largely disappeared.”

4) Fig. 4 is good, but missing a good placement. try to write more on it and put it before conclusion.

As suggested, we have expanded more on it and added before conclusion

“The high protein diet is the frequently subscribed diet post-EBTs. The protein-based bariatric plate model by Cambi et al. describes the macronutrient and micronutrient composition of daily meals to promote and maintain weight loss over the long term after EBT (Figure-5). The model recommends 50% calcium and iron-rich protein, 30% vitamins, minerals, and fibers, represented by fruits and vegetables, and 20% whole carbohydrates with a low glycemic index [37]. Lipids in the form of Canola or olive oil were suggested. The high protein and fibre rich diet are thought to be associated with improved satiation, satiety, and reduced food intake”

5) I have missed the supplementary data, I couldn't find it in this manuscript.

We have submitted all the files for review

Reviewer 2 Report

Summary of manuscript: This article discussed endoscopic bariatric therapies (EBT). The authors covered nutrition and management of obesity following EBT.

General comments: I carefully reviewed this manuscript. The authors provided an interesting manuscript, which is written in an understandable manner. I provided further comments below.   

Point 1: Line 1: The type of paper should be selected.

Introduction

Point 2: Line 36: Please change “lowers” to “lower”

Endoscopic Bariatric Therapies and Nutritional Care

Point 3: Line 71: Should “or” be changed to “of”?

Intragastric Balloons

Point 4: Line 77: Should “balloon” be plural?

Point 5: Line 86: Should “off” be changed to “of”?

Point 6: Line 87: Should “patients” be changed to “patient”?

Point 7: Figure 1: “A” and “B” are not indicated in the figure.

Endoscopic Gastroplasty

Point 8: Line 98: Should “reduces” be changed to “reduced”?

Point 9: Line 100: Should “raise” be changed to “rise”?

Point 10: Line 100: Should “prevent” be plural?

Point 11: Figure 2: “A” and “B” are not included in the figure.

Peri-procedure

Point 12: Line 135: Should “calorie” be changed to “calories”?

Point 13: Line 138: I am not able to locate Table 1.

Stage-1

Point 14: Line 144: “Kcal” is indicated here; however, on line 135, “kcal” is stated. Please be consistent throughout the manuscript.

Stage -3

Point 15: Line 160: There is an extra space in “Stage -3”

Point 16: Figure 3: Under “Plan in advance”, please capitalize “pack”. Please change “you” to “your”; under “Liquids”, please change to “throughout”.

Point 17: Figure 3 legend: Lines 178-179: Please provide the abbreviation definitions for EBT and MDT.

Healthy eating practice

Point 18: Line 193: Please change “carbohydrate diet” to “carbohydrate diets”

Point 19: Line 212: Should Figure 4 be cited here? Is reference 33 correct?

Point 20: Figure 4 legend: Line 262: Is reference 33 correct? Is “Cambri” spelled correctly? Is the figure original and not published elsewhere? Please be sure all citations and references are correct in the manuscript.

Point 21: Line 265: Please complete the “Author Contributions” section.

Point 22: Line 274: Please complete the “Institutional Review Board” statement and the remaining sections below.

Author Response

Prof. Maria Luz Fernandez

Editor-in-Chief

Nutrients Journal

Department of Nutritional Sciences,

University of Connecticut, Storrs, CT 06269, USA

Manuscript ID: nutrients-1804837

July 24, 2022

Dear Editor and Reviewers,

Thank you very much for the reviews and comments. We have amended the manuscript as suggested.

Reviewer 2

6) Introduction Point 2: Line 36: Please change “lowers” to “lower”

 We have changed as suggested.

7) Endoscopic Bariatric Therapies and Nutritional Care Point 3: Line 71: Should “or” be changed to “of”?

We have changed as recommended.

8) Intragastric Balloons Point 4: Line 77: Should “balloon” be plural?

We have changed to balloons.

9) Point 5: Line 86: Should “off” be changed to “of”?

We have edited to “of”.

10)Point 6: Line 87: Should “patients” be changed to “patient”?

We have changed as suggested.

11) Point 7: Figure 1: “A” and “B” are not indicated in the figure.

We have indicated the figure in the manuscript.

12) Endoscopic Gastroplasty Point 8: Line 98: Should “reduces” be changed to “reduced”?

We have changed to “reduced.”

13) Point 9: Line 100: Should “raise” be changed to “rise”?

We have amended it to “rise”.

14) Point 10: Line 100: Should “prevent” be plural?

We have changed as suggested.

15) Point 11: Figure 2: “A” and “B” are not included in the figure.

We have indicated in the manuscript.

16) Peri-procedure Point 12: Line 135: Should “calorie” be changed to “calories”?

We have changed it to calories.

17) Point 13: Line 138: I am not able to locate Table 1.

We have inserted all the tables at the end of the manuscript.

18) Stage-1 Point 14: Line 144: “Kcal” is indicated here; however, on line 135, “kcal” is stated. Please be consistent throughout the manuscript.

We have amended it.

19) Stage -3 Point 15: Line 160: There is an extra space in “Stage -3”

We have fixed it.

20) Point 16: Figure 3: Under “Plan in advance”, please capitalize “pack”. Please change “you” to “your”; under “Liquids”, please change to “throughout”.

We have changed the figure as per recommendations.

21) Point 17: Figure 3 legend: Lines 178-179: Please provide the abbreviation definitions for EBT and MDT.

We have elaborated on the abbreviation

22) Healthy eating practice Point 18: Line 193: Please change “carbohydrate diet” to “carbohydrate diets”

We have changed as suggested.

23) Point 19: Line 212: Should Figure 4 be cited here? Is reference 33 correct?

We have amended to read it correctly.

24) Point 20: Figure 4 legend: Line 262: Is reference 33 correct? Is “Cambri” spelled correctly? Is the figure original and not published elsewhere? Please be sure all citations and references are correct in the manuscript.

We have amended the spelling and the figures are original.

25) Point 21: Line 265: Please complete the “Author Contributions” section.

We added the contributions to the title page.

26) Point 22: Line 274: Please complete the “Institutional Review Board” statement and the remaining sections below.

We have added the Institutional board review statement to the title page.

Thank you again for the insightful comments to improve the manuscript.

Sincerely,

Dr. Anuradha Negi, MRCP, FAMS

Raffles Specialist Center

Singapore.

Round 2

Reviewer 1 Report

The authors have improved the manuscript, but i couldn't find supplementary files. why you have added tables in supplementary documents as a review manuscript/article. it would be better to add all the contents to single manuscript file, so that all the readers get full information in single file.

Author Response

Dear Reviewer,

Thank you very much for the comments. We have amended the word file as per your request. We have included all the files under one word document. There are no supplementary files

Thank you

Reviewer 2 Report

I would like to thank the authors for making revisions. I have further comments below.

Point 1: Line 1: The type of article should be indicated.

Point 2: Line 24. Please remove one of the periods after “EBT..”

Point 3: Figure 1: “A” and “B” are not indicated on the figure itself.

Point 4: Figure 2: “A” and “B” are not indicated on the figure itself.

Point 5: Line 153: Unfortunately, I am not able to locate Table 1.

Point 6: Figure 3: Under “Plan in advance”, please capitalize “pack”. Please change “you” to “your”; under “Liquids”, please change “though out” to “throughout”.

Point 7: Figure 3 legend: Lines 198-199: Please provide the abbreviation definitions for EBT and MDT.

Point 8: Lines 252-253: Please remove these two figure 4 legends, as the figure 4 legend is below figure 4.

Point 9: Line 255: Figure 4 legend: Please change “off” to “of”

Point 10: Lines 259-261: It is stated that the Mediterranean diet consists of a moderate intake of fish and dairy, and a low intake of red meat. However, figure 4 does not indicate these characteristics. Should the “allowed” icon be changed to “moderate”?

Point 11: Line 271: Should “Canola” be capitalized?

Author Response

Dear Reviewer,

Thank you for the comments. We have amended the manuscript to include all the changes recommended.

Point 1: Line 1: The type of article should be indicated.

We have indicated as review article in the title page.

Point 2: Line 24. Please remove one of the periods after “EBT..”

We have amended it

Point 3: Figure 1: “A” and “B” are not indicated on the figure itself.

We have indicated in the figure

Point 4: Figure 2: “A” and “B” are not indicated on the figure itself.

We have indicated in the figure

Point 5: Line 153: Unfortunately, I am not able to locate Table 1.

We have added both the table 1 and 2

Point 6: Figure 3: Under “Plan in advance”, please capitalize “pack”. Please change “you” to “your”; under “Liquids”, please change “though out” to “throughout”.

We have changed as suggested.

Point 7: Figure 3 legend: Lines 198-199: Please provide the abbreviation definitions for EBT and MDT.

We have expanded the abbreviation in the figure

Point 8: Lines 252-253: Please remove these two figure 4 legends, as the figure 4 legend is below figure 4.

We have removed it.

Point 9: Line 255: Figure 4 legend: Please change “off” to “of”

We have changed it as suggested.

Point 10: Lines 259-261: It is stated that the Mediterranean diet consists of a moderate intake of fish and dairy, and a low intake of red meat. However, figure 4 does not indicate these characteristics. Should the “allowed” icon be changed to “moderate”?

We have modified the table as suggested.

Point 11: Line 271: Should “Canola” be capitalized?

We have changed to "canola"

Thank you for the thorough and insightful comments for improving the manuscript.